behaviour/cognition/ecology

protogynous hermaphrodites, cognitive performance, learning, inhibitory control, fish, cleaning mutualism

**Author for correspondence:**
Zegni Triki
e-mail: zegni.triki@gmail.com

# Sex differences in the cognitive abilities of a sex-changing fish species *Labroides dimidiatus*

## Zegni Triki[1,2] and Redouan Bshary[2]

[1]Institute of Zoology, Stockholm University, Stockholm 106 91, Sweden
[2]Institute of Biology, University of Neuchâtel, Emile-Argand 11, 2000 Neuchâtel, Switzerland

 ZT, 0000-0001-5592-8963; RB, 0000-0001-7198-8472 

Males and females of the same species are known to differ at least in some cognitive domains, but such differences are not systematic across species. As a consequence, it remains unclear whether reported differences generally reflect adaptive adjustments to diverging selective pressures, or whether differences are mere side products of physiological differences necessary for reproduction. Here, we show that sex differences in cognition occur even in a sex-changing species, a protogynous hermaphroditic species where all males have previously been females. We tested male and female cleaner fish *Labroides dimidiatus* in four cognitive tasks to evaluate their learning and inhibitory control abilities first in an abstract presentation of the tasks, then in more ecologically relevant contexts. The results showed that males were better learners than females in the two learning tasks (i.e. reversal learning as an abstract task and a food quantity assessment task as an ecologically relevant task). Conversely, females showed enhanced abilities compared with males in the abstract inhibitory control task (i.e. detour task); but both sexes performed equally in the ecologically relevant inhibitory control task (i.e. 'audience effect' task). Hence, sex-changing species may offer unique opportunities to study proximate and/or ultimate causes underlying sex differences in cognitive abilities.

## 1. Introduction

For many gonochoric (i.e. non-sex changers) vertebrate species, it has been shown that males and females do partly differ with respect to their average cognitive performance, in certain tasks, in humans [1], monkeys [2], rats [3,4], birds [5] and fish [6]. Differences in brain development due to distinctive hormonal and neurohormonal pathways, which differ between sexes [7], might be the underlying mechanisms for both adaptive and non-adaptive

differences between sexes [8]. For instance, the role of gonadal steroid hormones (mainly testosterone) in promoting sex differences in spatial learning tasks has been studied thoroughly in several species. Male rats, for example, excel in spatial tasks compared with females as well as with males and females administered with an androgen blocker. Giving testosterone to female rats, on the other hand, improves performance [4].

Males and females' cognitive abilities can be subjected to different selection pressures as a consequence of sexual selection and diverging life histories. So, coupling brain development and resulting cognitive performance to such hormonal pathways' activity may yield adaptive differences [9–11]. Depending on the underlying genetics, some cognitive abilities might be the outcome of sexual conflict, where the expression of some alleles in a female may have the opposite effects than if expressed in a male [12]. In guppies, for instance, most evidence on sex differences in cognitive abilities tend to be explained by differences in the reproductive strategies of the two sexes, such that females, often driven by foraging motivations, have greater cognitive flexibility, while males, driven by finding mates, excel in spatial memory tasks (see review by Cummings [11]).

Different studies reported sex differences in specific cognitive performances and brain morphology that can indeed be attributed to selection based on different ecological needs. For example, the females of parasitic cowbirds have a larger hippocampus (a brain region linked to spatial navigation) [13] and perform better than males in spatial memory tasks [5]. These females rely on spatial memory to relocate nests of potential hosts in their environment, while males do not need such spatial memory skills in their lives [5]. Laboratory experiments on fish show that male guppies allowed to reproduce with a female partner had larger brains compared with those kept in the same-sex pairs [14]. Adaptive reasons for many other differences, however, remain obscure. In the best-studied example, many differences between male and female humans in IQ tasks are difficult to explain in a functional way [15]. Moreover, it is currently unclear to what extent selection on divergent (neuro)hormonal pathways in the context of reproductive roles cause such differences, either in an adaptive way or with cognitive differences being a mere side effect [15]. It is clear that hormones, such as testosterone, cannot be the only mechanism underlying sex differences in cognitive performance, as the same cognitive domain can be more advanced in either sex, depending on the species tested [16]. For example, experiments on reversal learning as a measure of cognitive flexibility revealed higher performance in females in some species, like in rats and guppies [6,17]; but for other species, like in great tits and zebra finches, it was the males who performed better [18,19].

To advance our understanding of the links between sex and cognitive performance with respect to mechanisms and function, while also considering potential constraints and sexual conflict, sequential hermaphroditism potentially offers a suitable study model. In such sexual systems, individuals reproduce first as one sex when they are young and small (i.e. male in protandry and female in protogyny), and then switch to reproducing as the opposite sex when they are old and big [20]. Therefore, the very same individual needs to solve the challenges specific to each sex during its life.

Here, we used a protogynous fish species, the cleaner wrasse (*Labroides dimidiatus*, hereafter 'cleaner'), as a study system. Cleaners are haremic group-living fish; individuals start their lives as females, and then only the largest and socially dominant individuals change sex to become male to defend and control a harem [21]. This yields to males being generally more aggressive than females [22]. Males regularly aggress female harem members, especially female(s) with a similar body size. This is because a similar-sized female poses a threat to the dominant male if it changes sex and becomes a male competitor [22,23], but receiving aggression inhibits sex change [22]. In the absence of the male, the largest female in the harem changes sex to a male [22]. There is no fixed age to change sex, but rather a socially controlled sex change coupled with hormonal and neurohormonal changes (for more details on the social and physiological mechanisms of sex change, see reviews by Godwin [24,25]).

Another interesting feature of cleaners' complex social life is their engagement in mutualistic cleaning interactions with a variety of coral reef fishes (hereafter 'client') [26]. Female cleaners occupy small territories called 'cleaning station' while male cleaners have a larger territory composed of several of these females' cleaning stations [21]. Given the accessibility options to a cleaning station, client fish can be categorized into two classes, either as 'resident' with access to a single cleaning station or as 'visitor' with access to multiple stations [27]. The cleaner–client social interactions are complex because cleaners prefer to cheat by eating client's mucus instead of cooperating by eating client's ectoparasites [28], and because visitor clients often swim away if made to wait for the cleaning service [27]. There is limited evidence with respect to sex differences in cleaning services. While females reduce their biting rate significantly towards (visitor) clients in the presence of a male partner, both sexes tend to bite at a similar rate when inspecting a client alone [29].

Besides, cleaners show evidence of great strategic sophistication, based on individual recognition of clients [30], categorization of client types [27,31], bookkeeping of past interactions [32], social competence

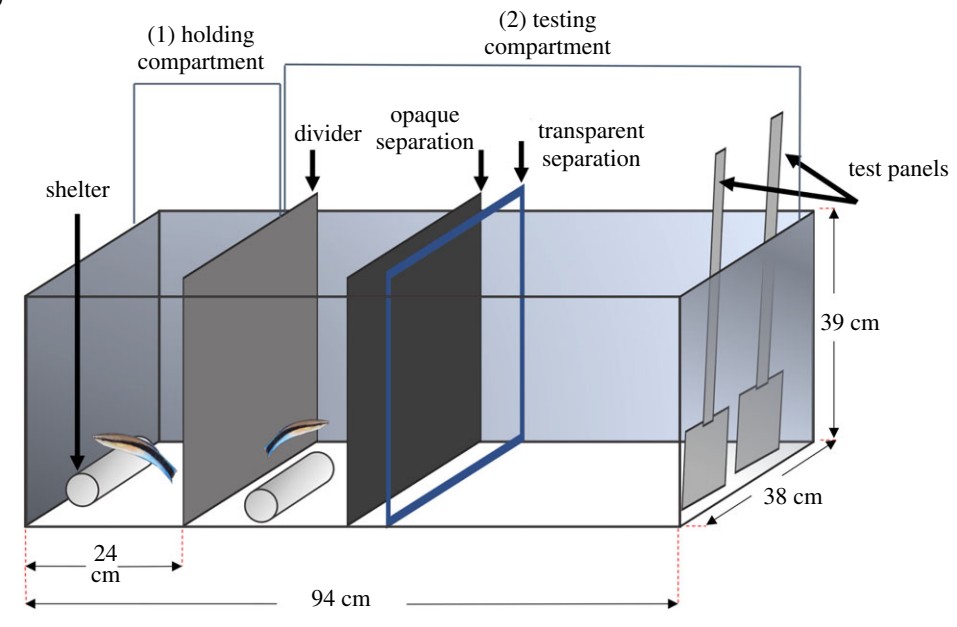

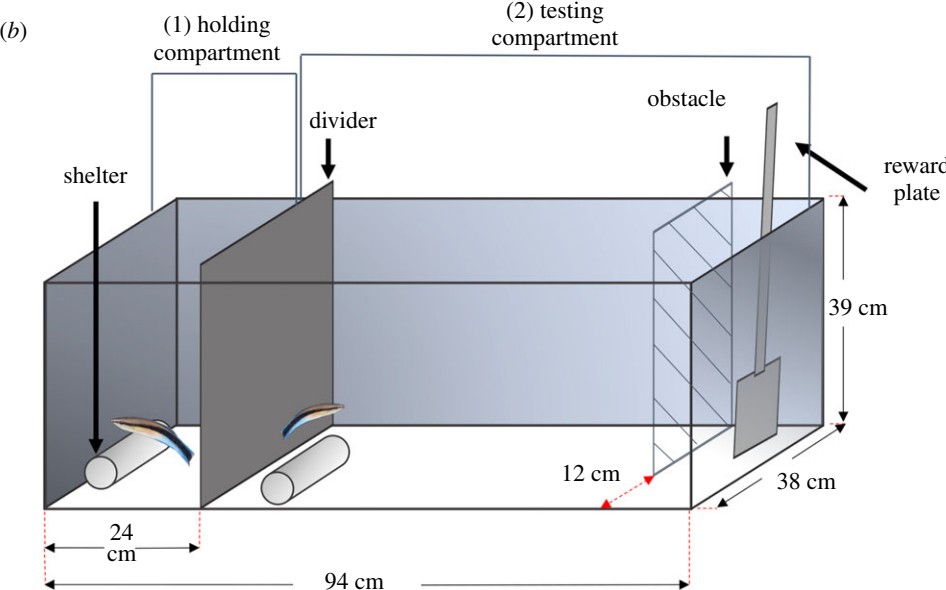

**Figure 1.** Schematic of the experimental aquaria. All fish were tested at their home aquaria. The divider in the aquarium helped to separate the male and female cleaner during trials. It created a holding (1) and testing (2) compartment. In this example, the female cleaner (i.e. the smaller fish) is in the testing compartment, while the male cleaner (i.e. larger fish) is in the holding compartment. The paradigm in (a) shows an example of two plates, like in the reversal-learning and 'client quality' tasks, as well as in the treatment condition of the audience effect task (i.e. the control was a single plate). The paradigm in (b) shows an example of the detour task. The opaque and transparent separations are shown in (a), and also used for the detour task, helped to isolate the focal individual and prepare the test set-up before each trial.

[33], reputation management and tactical deception [27]. Further studies have also documented a set of advanced cognitive abilities in cleaners, like generalized rule learning [34], numbering competence [35], long-term memory [36] and some mirror self-recognition [37]. So far, this evidence on cleaner fish complex cognition has been collected on females alone, mainly for practical reasons such that females are more abundant than males and easier to locate and capture.

Our aim was to understand whether there are potential sex differences in cleaner fish cognition. As any potential differences could be linked to differences in their cleaning ecology, we first recorded the natural cleaning behaviour of wild male and female cleaners. Then, we tested male and female cleaner fish in a battery of laboratory-based cognitive tasks (figure 1). The tasks tested the same

cognitive domain once in an abstract context, and once in ecologically relevant context [38]. We used experimental paradigms with an abstract representation of cues, like reversal learning and detour task, to target and test domain-general cognitive processes. On the other hand, paradigms that were more tightly linked to cleaning behaviour tested for specialized cognitive solutions to specific problems (i.e. domain-specific cognition) [38], i.e. the ability to prefer a more food-offering client over a less food-offering client, and to eat against preference in order to obtain overall more food. Testing different cognitive domains in individuals of both sexes should be the norm to understand better individual-related and sex-related variation in cognitive traits [39].

Labroides dimidiatus is a wrasse species with indeterminate growth; that is, the older they get, the larger they become. In other words, larger individuals (within or between sexes) are more likely to be older [40]. Previous research on cleaner fish has shown that performance levels from various cognitive tasks do not correlate with body size (a proxy for age) [41]. Therefore, it is unlikely that males will outperform females in any of our four cognitive tasks just because they are larger and older. Also, given that all males had been females earlier in their life, a reasonable null hypothesis would be that there are no differences in cognitive performance between the two sexes. The alternative hypothesis is that male and female cleaners will differ in their cognitive performance because their different intraspecific social roles warrant different cognitive abilities. For example, females could show both more self-restraint and flexibility as these abilities may help to manage social relationships in a size-based hierarchy. On the other hand, if cognition promotes growth and survival in the females, and thus sex-changing later in life, one would expect that males could potentially perform above female average. Thus, we do not have clear predictions on the direction of potential differences in male and female cognitive performances, especially that there is no clear evidence on male cleaner fish cognition compared with the extensive data on females. This study can hence be viewed as an explorative study rather than testing fixed hypotheses.

# 2. Methods

We conducted the present study at Lizard Island (14°66'82" S, 145°46'4" E), Great Barrier Reef, Australia, in June and July 2019. We first collected the behavioural data at a reef location called Clam Gardens (14°39'47.1" S, 145°27'01.3" E). Afterwards, we collected cleaners from this site to test them in four cognitive tasks at the Lizard Island Research Station (LIRS) facilities. At the end of the experiments, we returned and released all the caught cleaners at their respective site of capture.

## 2.1. Behavioural observations

We aimed at observing the behaviour of male and female pairs of cleaner fish in their natural habitat. Therefore, we randomly targeted eight males with their female partners from different harems, and video-recorded their behaviour for 30 min. We went scuba diving to record cleaner's behaviour from a distance of approximately 2 m to minimize disturbance. We recorded all videos between 08.30 and 16.00, using Canon G15® and GoPro Hero3® cameras. From these observations, we extracted the cleaner–client interaction patterns per cleaner and per client class (i.e. visitors and residents). Overall, we estimated the time spent by cleaners in cleaning interactions, the total number of interactions per time unit (i.e. 30 min) and the frequency of client jolts per 100 s of cleaning interactions.

## 2.2. Laboratory cognitive tasks

### 2.2.1. Animal capture and housing

To test cleaners' cognitive performance, we captured 10 established pairs of male and their female partner—those found together with the males at the time of capture—from Clam Gardens at Lizard Island. To do so, we used a barrier net (2 × 1 m, 5 mm mesh size) and hand nets. Afterwards, we transported the captured fish in identified zip plastic bags filled with aerated seawater to the facilities of LIRS, and housed every pair in a glass aquarium measuring 39 × 94 × 38 cm (height × length×width); provided with shelters (PVC tubes: 10 × 1 cm). The 10 collected males measured (mean ± s.d.) 8.49 ± 0.36 cm body total length (TL), and 5.10 ± 0.63 g body mass, while the 10 females measured 7.35 ± 0.51 cm TL, and weighed 3.28 ± 0.65 g. We allowed the fish to acclimatize for 14 days prior to starting the cognitive experiments. Meanwhile, we daily fed them with mashed prawn

smeared on Plexiglas plates measuring $15 \times 8$ cm (length × width). During the cognitive tasks, fish acquired food solely from the trials, and no further feeding was performed.

For the sample size ($N = 10$ pairs), we aimed at using the minimum number of subjects that can allow us to carry out meaningful statistical analyses. Using larger numbers of wild animals is not easy to achieve compared with laboratory-based animals. Ethics guidelines [42] often favour minimum samples. Also, given that our study system is female-biased, removing 10 males must have induced some potential disturbances on the sites of capture, wherein these males have lost their harems possibly to other male competitors or to the next dominant female.

We tested cleaners in their house aquaria individually. To do so, we divided the house aquaria into two compartments: holding compartment and testing compartment. We allowed fish to acclimatize for at least 10 min after introducing the opaque divider in the aquaria before the start of trials. During the trials, we used two partitions (i.e. one opaque and one transparent) to confine the focal individual to one side of the test compartment. Meanwhile, we placed the test plate(s) on the other side of the test compartment that was inaccessible by the cleaner. Once the test plate(s) were ready, we removed the opaque partition followed by the transparent one. This allowed the fish to see the test plate(s) before granting it access (figure 1). It is important to note here that in all the cognitive tasks we perform in this study, we used Plexiglas plates with food as surrogates for client fish. The tasks did not involve 'real' clients at any stage.

It is noteworthy to mention that the order of testing first either the females or the males was counterbalanced. Fish were tested from 8.00 to 12.00 and from 13.00 to 17.00. A break of 1 h separated the morning session from the afternoon one, wherein the female and male of every pair were allowed to join each other. Finally, between every two cognitive tasks, all cleaners were allowed 1 day of rest.

### 2.2.2. Reversal-learning task

In preparation for the reversal-learning task, cleaners had to learn an initial task first (associative learning). In this acquisition phase, we presented the cleaners with two novel Plexiglas plates of identical size ($8 \times 5$ cm, length × width), but of different colours, one yellow and one red. The yellow plate was always the one with a food reward at the back (i.e. two food items of mashed prawn), while the red plate had no food reward. To facilitate the learning of the acquisition phase, we first allowed cleaners to explore the test paradigm for two trials as pairs, then for five trials as individuals. During this pre-trial phase, fish were allowed to explore both the yellow and the red plates. On the other hand, once we started the trials *per se*, choosing the wrong plate resulted in the withdrawal of both plates from the aquarium. Upon solving the initial test, we then tested cleaners' ability in solving a reversal version of the task wherein we reversed the plates' roles: the red plate became the rewarding plate instead of the yellow one. The location of the rewarding plate (i.e. either right or left) followed a random sequence to ensure that the cleaners learned the colour cue of the plate and not its location in the aquarium. Cleaners were daily tested in two sessions, wherein each session consisted of 10 consecutive trials. The success criterion was set at either three consecutive seven correct choices per session, two consecutive eight correct choices per session or a single 9 or 10 correct choices per session. All fish solved the initial phase within 40 trials, and they were allowed up to 100 trials to solve the reversal-learning task. In this task, we compared cleaners' performance based on the number of trials needed to solve the reversal-learning task. One male did not reach the learning criterion in the acquisition phase and was not tested in the reversal phase.

### 2.2.3. Detour task

The detour task consisted of placing a novel Plexiglas plate measuring $8 \times 5$ cm (length × width) of plain grey colour behind a see-through obstacle. The plate offered a food reward (i.e. one item of mashed prawn) placed on its front side. The see-through obstacle was a transparent barrier measuring $26 \times 39 \times 0.2$ cm (width × length × thickness) on which we have drawn diagonal lines with 4 cm spacing. By placing the obstacle parallel to the opaque divider (figure 1), either on the left or the right, we created an opening of 12 cm to access the reward plate. The position of the opening was randomized through trials, with 50% of the time being on the right side and 50% on the left side. We again offered a pre-trial of acclimatization, wherein pairs explored and familiarized with the experimental paradigm before the task. On the next day, we tested cleaners in two sessions of 10 trials each. We scored cleaners' performance as either pass or failure during each trial. Cleaners needed to inhibit their motor impulses to reach for the food reward directly and bump into the barrier but instead had to move away from the goal to reach it ultimately. That is, a pass consisted of swimming around the obstacle without touching it

to access the food reward. A failure, however, was whenever a cleaner bumped into the obstacle before reaching for the food reward (adapted from MacLean *et al.* [43]). We then estimated cleaners' performance in this task as the proportion of trials without barrier touching from total trials.

### 2.2.4. Client quality task

In the 'client quality' task, we used two novel Plexiglas plates of similar size (length × width; 10 × 7 cm) but of different colours and patterns, such as vertical pink stripes or horizontal green stripes to facilitate the visual distinction between the plates by cleaners. Here, we placed the food reward on the back of the plates where one plate offered two items of mashed prawn while the other plate offered only one prawn item. Furthermore, we assigned five pairs to have the green plate as the highly rewarding plate, while the other five had the pink plate as the highly rewarding one. The optimal strategy was to choose a highly rewarding plate. Eating off either rewarding plate led to the immediate withdrawal of the other plate from the aquarium. We tested cleaners daily in two sessions of 10 trials each. Overall, we allowed cleaners a maximum of 100 trials to solve the task. The learning criterion was the same as the criterion set for the reversal-learning task: three consecutive 7/10, two 8/10 or a single 9/10 correct choices. Since several cleaners failed the task within 100 trials, we evaluated their performance as either success or failure.

### 2.2.5. Audience effect task

In nature, while cleaners are interacting with a current client, they are often surrounded by potential future clients (i.e. bystanders). Although cleaners prefer to bite client's mucus instead of eating ectoparasites which constitute cheating [28], they can adjust their feeding preferences in the presence of bystanders and refrain from biting the current client [44]. It is possible to reproduce this natural situation of the audience effect in laboratory settings by substituting real clients with Plexiglas plates and high- and low-preferred types of food with mashed prawn and fish flakes, respectively [44]. For this reason, we first subjected cleaners to training trials where they learn to feed off Plexiglas plate offering 12 flakes items and two prawn items. The consumption of a highly preferred food item resulted in the immediate withdrawal of the feeding plate(s) while eating a low-preferred food item(s), however, had no consequences. In total, we ran three 'feeding against the preference' training trials in 1 day.

During the task trials, we used two novel grey Plexiglas plates (length × width, 12 × 7 cm) with either yellow or white stripes as decoration. Every plate offered four food items in total: two flakes items and two prawn items. Following the same logic during training, the plates would remain in the aquarium as long as the fish are eating flakes items only. Eating a prawn item, however, would lead to the withdrawal of both plates. The optimal option in this task was to eat all flakes item available and then eat a prawn item. The trials with two plates represent the treatment part of the experiment: whichever plate the cleaners are feeding on first, makes the second plate a 'bystander' plate. For instance, cleaners can have access to the 'bystander' plate as long as they eat only flakes off the first plate. We also ran control trials with a single plate (i.e. absence of 'bystander' plate). Here, we used one of the two plates (i.e. a counterbalance between a plate with yellow stripes and the one with white stripes) offering two flakes and two prawn items. We ran control and treatment trials in multiple rounds, where each round was composed of one treatment trial and one control trial. We allowed 30 min interval between every two trials, and subsequently 60 min interval between every two rounds. We flipped a coin to determine the order of the two trials within each round. In total, we ran six rounds over 2 days.

During every trial, we recorded the number of flakes items and prawn items eaten by the cleaners. From there, we averaged cleaners' performance per plate role (i.e. first plate from treatment, and single plate from control) through the rounds. We then estimated the ratio of flakes items to prawn items eaten per plate role.

## 2.3. Data analyses

We ran the statistical analyses and generated figures with the open-source statistical software R v. 3.6.2 [45]. Since cleaners belonged to identified pairs in either the behavioural observations or in the cognitive tasks, we hence used pair identity as a random intercept in our statistical models. To analyse the data, we fitted a set of Bayesian linear mixed-effects models (blmer) and Bayesian generalized linear mixed-effects models (bglmer) from the R package `blme` in R language. The syntax we employed in every statistical model was as follows: dependent variable ~ sex + (1 | pair identity). We fitted the sex of cleaners as a

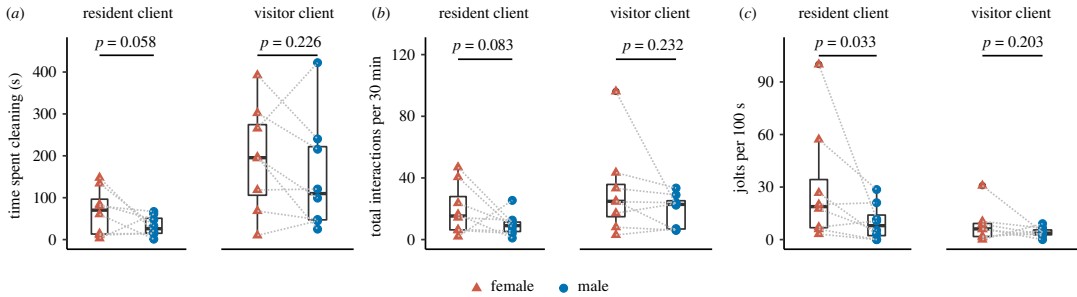

**Figure 2.** Cleaner–client natural interactions. Boxplots, median and interquartile of (*a*) the overall cleaning duration in 30 min video observations shown per client class, (*b*) the total number of interactions in 30 min video observations and (*c*) frequency of client jolt scaled to 100 s of cleaning interactions. *p*-values were from the statistical analyses (refer to the main text for further details). Coloured circles and triangles are the raw data points, while grey dashed lines connect the female and male of each pair.

fixed factor while cleaners' behaviour (i.e. time spent cleaning, total interaction and jolt rates with visitors and residents) and cognitive performance (i.e. trials to solve the reversal learning, proportion of correct detours and prawn to flake ratio consumed in the 'audience effect' task) as dependent variables. For the model testing 'audience effect', however, treatment (i.e. control versus treatment plate) was added to the model as a predictor.

Furthermore, we estimated the explained variance (marginal and conditional $R^2$) for linear models and pseudo-marginal and conditional $R^2$ for the nonlinear models, 95% confidence intervals and effect size for continuous variables (i.e. as Cohen's *d* coefficient: [mean males – mean females]/ standard deviation). For *post hoc* analyses, we used the function `emmeans()` from the `emmeans` package in R language. We also checked and tested models' assumptions, like the normality of the residuals' distribution and homogeneity of the variance, via statistical models and visual plots. To check for a potential link between body size (also a proxy for age) and performance within each sex, we generated a correlation matrix of performance from every task, body weight and body length for the tested male and female cleaners. A detailed code showing step-by-step the data analyses is accessible in a public data repository along with the dataset (see Data accessibility statement).

# 3. Results

## 3.1. Behavioural observations: cleaner–client interactions

The cleaner–client behavioural aspects, like time spent cleaning (figure 2*a*) and interaction frequency, did not significantly differ between females and males (figure 2*b* and table 1). The analyses, however, showed a significant main sex difference in residents' jolt rate (i.e. a proxy for cleaner biting) (figure 2*c*). That is, female cleaner significantly cheated more their resident clients, by taking mucus bites instead of removing ectoparasites, which triggered residents' jolt reaction. Figure 2*c* shows that there is a potential outlier in the female data. Even with removing this outlier in the analysis, we still found that females bit resident clients more frequently than the males (Bayesian LMER: $\chi^2 = 5.912$, d.f. = 1, $p = 0.015$, 95% CI [−17.7, −1.90]). Visitors' jolt rate, on the other hand, was not significantly different between the two sexes (figure 2*c* and table 1).

## 3.2. Cognitive performance

### 3.2.1. Learning abilities

In the reversal learning task, all fish solved the initial phase within 40 trials. Although all the tested females and males successfully learned the reversal, the males were significantly faster to reach the learning criterion than females (figure 3*a* and table 1). In the 'client quality' task, all male cleaners solved the task within 100 trials, but only four out of 10 females solved the task. The differences between the two sexes' performance were statistically significant (figure 3*b* and table 1).

In the detour task, it is noteworthy to say that all individuals solved the task in the sense that they accessed the food eventually. With a strict criterion for correct performance (i.e. detouring without touching the barrier [6,43,46]), females and males scored 36% and 22% correct detours, respectively.

**Table 1.** A summary of the statistical outcomes. Variables and values in bold refer to statistically significant differences ($\alpha$ set at $p \leq 0.05$). 95% CI, confidence interval. Marginal $R^2$, the proportion of variance explained by the fixed factor of the model, which was the sex (i.e. female versus male). Conditional $R^2$, the proportion of variance explained by the fixed and random factors; fixed factor was sex and random factor was the identity of the pair. Cohen's $d$ effect size: 0.2, small; 0.5, medium; greater than or equal to 0.8, large effect. F, females; M, males.

| category | model class | tested variable | | N | $\chi^2$ | p-value | marginal $R^2$ | conditional $R^2$ | 95% CI lower | 95% CI upper | effect size (Cohen's d) | F versus M | figure |
|---|---|---|---|---|---|---|---|---|---|---|---|---|---|
| behaviour | Gaussian | time spent in cleaning resident clients | sex | 8 F and 8 M | 3.588 | 0.058 | 0.129 | 0.463 | −70.1 | 1.19 | 0.81 | F ≥ M | figure 2a |
| | Gaussian | time spent in cleaning visitor clients | sex | 8 F and 8 M | 1.466 | 0.226 | 0.021 | 0.782 | −109 | 25.70 | 0.32 | F = M | figure 2a |
| | Gaussian | total interactions with resident clients | sex | 8 F and 8 M | 3.000 | 0.083 | 0.118 | 0.410 | −20.9 | 1.29 | 0.76 | F = M | figure 2b |
| | Gaussian | total interactions with visitor clients | sex | 8 F and 8 M | 1.428 | 0.232 | 0.023 | 0.758 | −0.80 | 0.19 | 0.56 | F = M | figure 2b |
| | Gaussian | **jolts rate by resident clients** | **sex** | **8 F and 8 M** | **4.518** | **0.033** | **0.125** | **0.585** | **−37.8** | **−1.53** | **0.80** | **F > M** | figure 2c |
| | Gaussian | jolts rate by visitor clients | sex | 8 F and 8 M | 1.617 | 0.203 | 0.065 | 0.393 | −10.2 | 2.17 | 0.55 | F = M | figure 2c |
| cognition / learning | Poisson | **reversal-learning task** | **sex** | **10 F and 9 M** | **13.067** | **<0.001** | **0.155** | **0.794** | **−0.35** | **−0.10** | — | **F < M** | figure 3a |
| abilities | binomial | **client quality task** | **sex** | **10 F and 10 M** | **5.519** | **0.019** | **0.411** | **0.631** | **0.79** | **6.68** | — | **F < M** | figure 3b |
| inhibitory control | Gaussian | **detour task** | **sex** | **10 F and 10 M** | **14.476** | **<0.001** | **0.157** | **0.794** | **−0.20** | **−0.07** | **0.91** | **F > M** | figure 4a |
| abilities | Gaussian | audience effect task | sex | 10 F and 10 M | 0.122 | 0.727 | 0.050 | 0.282 | −0.26 | 0.26 | 0.09 | F = M | figure 4b |
| | | | audience | 10 F and 10 M | 2.475 | 0.116 | | | −0.38 | 0.15 | 0.46 | — | figure 4b |
| | | | sex × audience treatment | 10 F and 10 M | 0.122 | 0.727 | | | −0.44 | 0.31 | — | F = M | figure 4b |

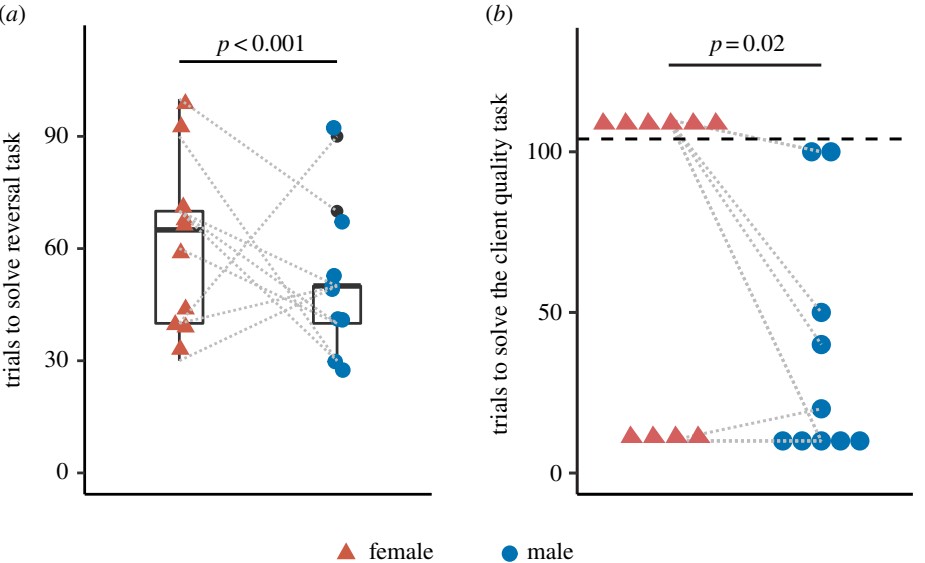

**Figure 3.** Learning abilities in male and female cleaner fish. (*a*) Boxplots, median and interquartile of the number of trials needed per cleaner to reach the learning criterion and solve the reversal-learning task. (*b*) Scatterplot of the number of trials needed per cleaner to reach the learning criterion and solve the 'client quality' task. The data points above the horizontal black dashed line refer to the individuals that failed to solve the task within the maximum allowed number of trials (i.e. 100 trials). *p*-values were from the statistical analyses (refer to the main text for further details). Coloured circles and triangles are the raw data points, while grey dashed lines connect the female and male of each pair.

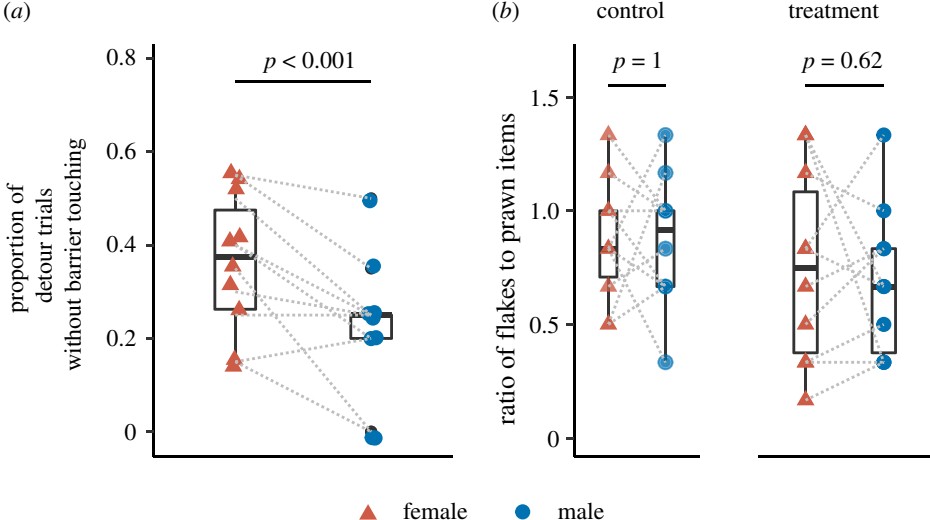

**Figure 4.** Inhibitory control abilities in male and female cleaner fish. Boxplots, median and interquartile of (*a*) the proportion of the trials, wherein fish detoured the barrier without touching it, calculated from a total of 20 trials in the detour task; and of (*b*) the ratio of flakes to prawn consumed in the control (i.e. single plate) or treatment (i.e. presence of an audience) conditions in the audience effect task. *p*-values were from the statistical analyses (refer to the main text for further details). Coloured circles and triangles are the raw data points, while grey dashed lines connect the female and male of each pair.

By comparing their performance, we found that the females significantly outperformed the males in this task (figure 4*a* and table 1).

### 3.2.2. Inhibitory control abilities

In the 'audience effect' task, the data analyses showed that there were no differences between females and males' feeding decisions (figure 4*b* and table 1), wherein both sexes ate against preference to a similar degree (*post hoc emmeans* test: female control versus male control, estimate = 0, *p* = 1.00; female treatment versus male treatment, estimate = 0.067, *p* = 0.625). Furthermore, it is important to say that

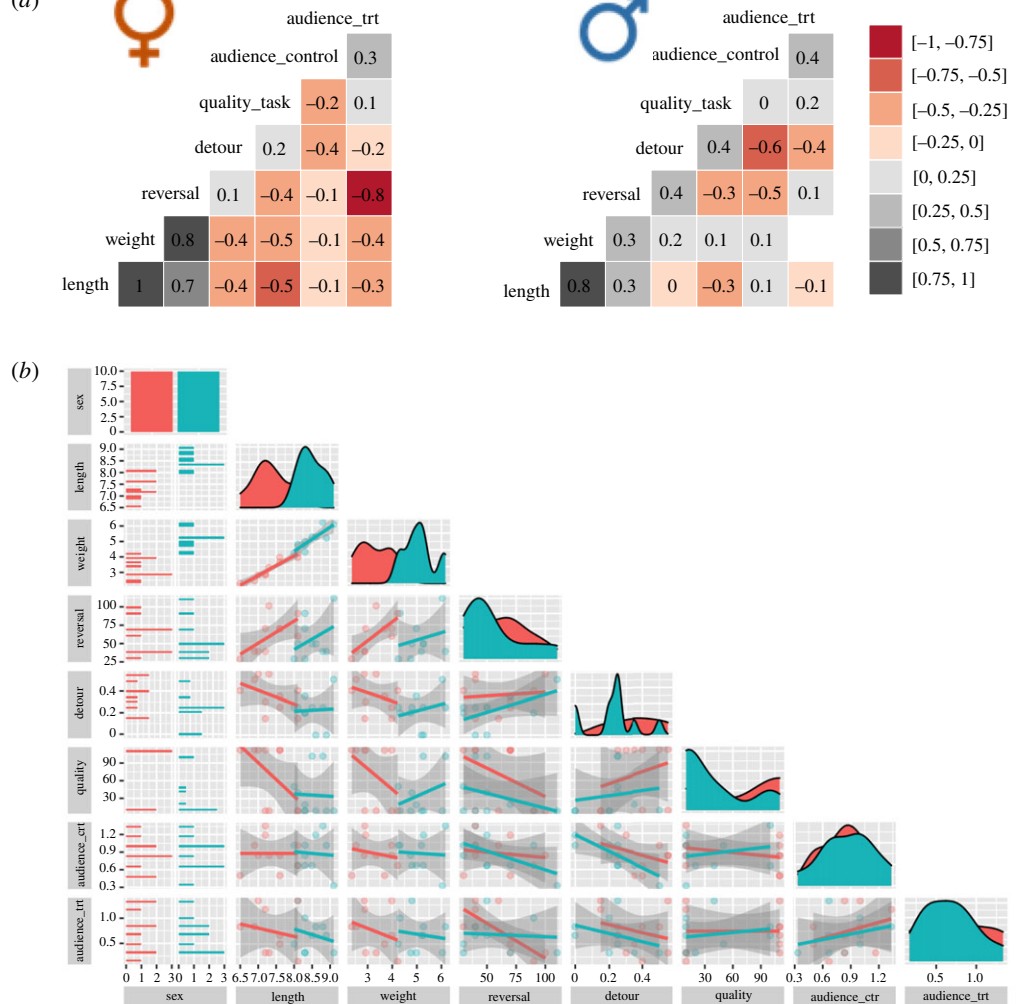

**Figure 5.** Correlation matrices of body measurements and performance in 10 male and 10 female cleaner fish. (*a*) Spearman correlation coefficients (meaningful coefficient for small sample size was set at $r \geq 0.7$). (*b*) Complementary visualization of data distribution and linear regressions of the variables from (*a*). 'Length' refers to body total length in cm; 'weight' is body mass in gram; 'reversal' is the number of trials needed to solve the reversal-learning task (i.e. low values means better performance); 'detour' is the proportion of correct detours; 'quality' task is the number of trials needed to solve the 'client quality' task (i.e. low values means better performance); 'audience_ ctr' indicates the flake to prawn ratio eaten by cleaners in the control condition of the audience effect task; 'audience_trt' indicates the flake to prawn ratio eaten by cleaners in the treatment condition of the audience effect task.

neither females nor males adjusted their feeding preferences to the presence of an 'audience' (*post hoc emmeans* test: female control versus female treatment, estimate = 0.117, *p* = 0.394; male control versus male treatment, estimate = 0.185, *p* = 0.185).

### 3.2.3. Correlations

In exploring whether body size (also a proxy for age) correlates with cognitive performance in males and females separately, we found a single significant relationship. In females, the number of trials needed to solve the reversal-learning task correlated positively with body size (Spearman's correlation coefficients: *N* = 10, *r* = 0.7 and *r* = 0.8 for TL and body weight, respectively); that is, learning the task takes longer when the females get bigger (figure 5).

## 4. Discussion

We had asked whether protogynous hermaphrodites show sex differences in their cognitive abilities. The key finding was that males showed more advanced learning abilities than females in both domain-

general and domain-specific representation of a learning task. Females showed better inhibitory control abilities in a domain-general task (i.e. detour task) than males. However, they failed to outperform males by applying these inhibitory control abilities to optimize their food intake in a more ecologically relevant context, i.e. feeding against preference ('audience effect' task). To our knowledge, the findings provide the first evidence of sex-related variation in the cognitive abilities of a protogynous species.

It is unlikely that males outperformed females in the learning tasks because they were larger and older. If that is the case, one would expect to find a positive relationship between high performance and body size within each sex (i.e. a proxy for age in fish with indeterminate growth [40]). As such, larger females (older) should outperform smaller ones (younger), and the same logic would apply to males. If anything, the opposite was true, where females' body size correlated positively with slow learning in the reversal task. Nevertheless, the males did outperform the females in this task despite being larger in body size. Performance in the other tasks, in both males and females, were independent of body size; a finding that fits previous outcomes resulting from a larger dataset on female cleaner fish [41]. It is noteworthy to mention that the observed and tested female cleaners were sexually mature, which means they were at least 1 year old (i.e. cleaners have a maximum life expectancy of 5 years [47]). Therefore, female cleaners must have had at least several hundred thousand interactions with clients of various quality (i.e. variable body size, mucus quality and ectoparasites load), before they were video-recorded and brought to the laboratory. This suggests that any differences between males and females with respect to cleaner–client interactions are quantitative and not qualitative.

## 4.1. Cleaning behaviour

The natural cleaner–client behaviour of the male and female cleaners served as an ecological background to understanding their performance in the laboratory-based cognitive tasks. Both males and females had similar cleaning interaction patterns, with main differences found in the biting rate of resident clients. For visitors, however, females and males cheated this client class at similar rates, which is in line with previous observations of cleaner fish in the Red Sea [29]. Together, these observations support the notion that females had plenty of experience in nature before being subjected to laboratory experiments. As such, any superior performance by males in the 'client quality' task (discussed below) cannot be explained by females lacking experience in interacting with various clients of different qualities with variable behavioural strategies towards cheating, like chasing the cleaner or switching to another one [48].

## 4.2. Cognitive performance

The null hypothesis that there are no differences between male and female cleaners' cognition was disregarded by our results. Two alternative hypotheses need to be hence considered. First, a change in hormonal profiles causes the observed differences in cognitive performance. At least in rats, female androgenization and male castration lead to an inverse effect on their cognitive performance [17]. If that is the case with cleaner fish, a follow-up question would be whether the changes in cognitive performance are adaptive or an unavoidable by-product of sex roles. We consider the latter possibility unlikely as females are not inferior learners in many gonochoric species [5,6,15,17]. If anything, females often excel in learning tasks compared with males, like in guppies [11].

Fish brains are highly plastic [14], and hence males could potentially adjust their brain structures and functions as a response to new selective pressures, like patrolling, defending and controlling a harem. Alternatively, better learning is always advantageous, but investment in relevant brain parts can only be increased once an individual changes sex because males can reduce investment in gamete production and use the available resources for brain growth instead. However, differential investment in reproduction is not a unique feature of sex-changing species [20], so the trade-off should cause males to be better learners in all species, which is not the case. A second hypothesis regarding our results is that better female learners are more likely to grow fast and survive, which makes them more likely to change sex and become males. As a consequence, male learning abilities would be better than the average female ability. Given the mismatch between the physiology hypothesis and available data, we currently consider it most parsimonious that males are former females that were above-average learners with higher flexibility. This is in line with previous findings on another sex-changing fish species (*Parapercis cylindrica*), where aggressive females become aggressive males [49].

Both the detour task and 'audience effect' task require some levels of inhibition to pass the tests. The former task is considered to be a part of the general-intelligence test battery [38], while the latter taps into

the ecology of the species where cleaners should refrain from eating high-preferred food rather than low-preferred food in order to increase their overall food intake [28,44]. The result that males performed poorer than females in the domain-general test (detouring without touching the barrier [43,46]) could be explained by field observations and laboratory tests showing that female cleaners cheat more often when alone than when a male cleaner is present [29]. In other words, females show a great deal of circumstantial inhibition of their food preferences compared with males. The abstract representation of self-inhibition task in the form of a detour task may have indeed captured this ecological difference between the two sexes.

The general failure of both males and females to adjust their feeding preferences in the presence of an 'audience' plate (i.e. a surrogate for audience client) has become rather the standard in recent years, apparently linked to a reduction in cleaner densities after major environmental perturbations [50,51]. Clients became more willing to queue for service [27,50], suggesting that they may also use less 'image-scoring' of cleaners' services as bystanders in nature. This change in client decision rules would have reduced the need for cleaners to show 'audience effect' in nature, which then could have caused them to fail in the experimental task (see discussion in [27]). As the Plexiglas plates simulated the behaviour of visitor clients, withdrawal upon consumption of a highly preferred food item (i.e. a prawn item as the equivalent of client mucus), the similar performance of male and female cleaners in the 'audience effect' task was consistent with the natural behavioural observations, wherein both sexes cheated visitor clients at similar frequencies.

The overall results show that not all cognitive domains are positively linked to one sex over the other. This insight provides avenues for studying the potential presence of sexual conflict [12] regarding cognitive abilities within the same individual first as a female, and then once it changes sex to become a male. In the first step, it needs to be established whether the degree of inhibitory control and/or learning abilities are stable over an individual's lifetime. If either were shown to be the case, a sexual conflict would arise if a cognitive function is positively associated with a higher reproductive output in one sex but negatively in the other.

# 5. Concluding remarks

Our study suggests a new avenue for research to address sex-related cognitive variation and its mechanisms in sequential hermaphrodite species. It also provides rare evidence of the links between ecologically relevant and domain-general learning abilities [38] in a sex-changing species. Kazancıoğlu & Alonzo [52] propose that protogynous hermaphroditism evolution in the Labridae fish family was driven by the male fitness increasing with body size (i.e. the size advantage hypothesis). This aspect needs further investigation to see whether sex change causes changes in cognitive performance or whether females with higher cognitive performance are those who are more likely to achieve sex change in their life. Promising future research will be hence evaluating the cognitive abilities of the very same individuals once as females and once when they change sex to males.

Ethics. The Animal Ethics Committee of the Queensland government (DAFF) approved the project under the number (CA 2019/06/1285).

Data accessibility. Data used in the study, the codes for statistical analyses and generating figures are available in the Figshare data repository: https://doi.org/10.6084/m9.figshare.8971088.

Authors' contributions. Z.T. and R.B. designed the study and collected the data. Z.T. analysed the data and wrote the first draft. Z.T. and R.B. finalized the paper.

Competing interests. The authors declare no competing interests.

Funding. Funding was provided by the Swiss National Science Foundation (grant no. 310030B_173334/1 to R.B.), and the Swiss National Science Foundation Early Postdoc Mobility grant (grant no. P2NEP3_188240 to Z.T.).

Acknowledgements. We kindly thank the staff of Lizard Island Research Station and Y. Emery for their field support. We thank Niclas Kolm for valuable comments on an earlier version of the manuscript.

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
