## [Peer Review File · Royal Society Open Science]

Review History

RSOS-210239.R0 (Original submission)

Review form: Reviewer 1 (Chiara Benvenuto)

Is the manuscript scientifically sound in its present form?

Yes

Are the interpretations and conclusions justified by the results?

No

Is the language acceptable?

Yes

Do you have any ethical concerns with this paper?

No

Have you any concerns about statistical analyses in this paper?

No

Recommendation?

Accept with minor revision (please list in comments)

Comments to the Author(s)

I was one of the previous referees and I really appreciated how the authors took on board the comments provided. The authors know already how I appreciate their choice of a study system (a sex-changing fish to look at differences in sex-specific cognition capability) and their smart experiments under lab conditions.

I really like the new supplementary Figure S1, which I would add in the main text, as it is very relevant to reduce the concern of differences in cognitive abilities due to age (with size as a proxy) because of extra experience, and not to sex.

I was worried for the low sample size, but the authors responded to my concerns (to be honest, I would include a shorter version of that answer in the material and methods, to explain the ethical implication and the sex ratio skew of removing more than 10 animals from the reef),

My other concerns/suggestions have been considered.

I am still unsure why the authors write: "On the other hand, as males are successful former females, they could potentially perform above female average" Why? In what tasks? I do not see the rationale: I thought the all point was to test for inter-sexual differences in performing various cognitive tasks. I don't understand where this general statement now comes from. And why one sex should in general perform better or worse (the authors mention that the different social roles warrant different cognitive abilities).

Re-reading the MS now, I have also a fundamental question. The authors start the paper aiming to look for potential sex differences in cleaner fish cognition. But then lots of speculation is on changing sex based on cognition. It seems that the focus shifts during the course of the MS, and I am not sure the new hypothesis on sex change is really tested with the clever lab experiments performed.

I do not understand why better learners are considered more likely to change sex. According to the size advantage model, as reported in the final discussion, larger female can become the dominant males. Are the authors suggesting that a smaller female, better learner (but again, in what tasks?) than a larger female, is more likely to change sex? And are they suggesting than this smaller female would grow faster (as males are bigger than the larger female)?

If males and females excel in different task, why is the most parsimonious hypothesis that males are former females that were above-average learners? The fact that aggressive females become aggressive males (which I brought to the attention of the authors in the previous review), just seems to show that the trait is maintained. Better learner females could be better learner males, but why they would be the ones preferentially changing sex? I really thought that size (and aggression) were important, to rule the harem. But maybe I am getting this wrong. Anyway, I am not sure the experiments performed support this hypothesis, as we do not know who is changing sex (and in some cases larger females perform worse than smaller ones, even though I could not explain why and the authors comment on it, but cannot explain it either).

Indeed, the authors state in line 528-530: "The overall results show that not all cognitive domains are positively linked to an individual's probability to change sex". But why not all? And where they get the probability from? Do they find evidence for some domains? They have not tested this directly. And in the final conclusion the authors state: "more investigation is needed to see whether sex-change causes changes in cognitive performance or whether females with higher cognitive performance are those who achieve sex-change later in their life" Should not be earlier in their life, if they preferentially change sex? But I really thought the whole MS was written based on the first consideration (sex change cause changes in cognitive performance).

I am just a bit confused and I think that these two need to be better clarified.

Finally, I don't know if I would talk about intersexual conflicts... Would not be the all point that hormones would influence some abilities, so changing hormones would change them too and

hormones would possibly amplify the effects on the cognitive abilities more relevant for that sex? I would not say that one cognitive ability is really detrimental to one sex: I would say it could be positive for both sexes, possibly more so in one sex than the other. But on this I am not sure.

Minor comments

Line 44: Now the introduction reads “males and females may partly differ”. I made a suggestion in the previous version and I still would go for it (but it is writing style, not important: it has been suggested that may or been shown they do.

Line 57: what pathways activities?

Line 62: consider revising punctuation: reproductive strategies of the two sexes, such that females, often driven by foraging motivations, have greater cognitive flexibility, while males, driven

by finding mates, excel in spatial memory tasks

line 91: I would remove -first in both male-first and female-first) as you mention that individual reproduce first as one sex (male or female), not male-first and female-first

Line 11: sorry, I am not familiar: do males have larger territories, and move from one cleaning station to the other? Are female cleaning stations inside the male ones?

Line 147: I would change cleaner fish into *Labroides dimidiatus*

Line 156: why intraspecific?

Line 161-162: change semicolon to comma, especially since there is

Line 172: just out of curiosity, how do you think males ill do when returned to the reef?

Line 195: TL in bracket

Line 196: I would add (mean \pm SD) the first time you see it

Line 361-362: would it be worth adding here (or in the discussion) that male do perform better (even if they are bigger)

Review form: Reviewer 2

Is the manuscript scientifically sound in its present form?

Yes

Are the interpretations and conclusions justified by the results?

Yes

Is the language acceptable?

Yes

Do you have any ethical concerns with this paper?

No

Have you any concerns about statistical analyses in this paper?

No

Recommendation?

Accept with minor revision (please list in comments)

Comments to the Author(s)

This study takes the examination of sex differences in cognition to a new level by comparing cognitive performance between the sexes in a species that has the ability to change sex. The authors conduct a series of cognitive assays, some that examine general cognitive features (such as cognitive flexibility and inhibitory control using detour task and reversal learning assays) and others that ask an ecologically relevant cognitive task (two foraging tasks that examined an

individual's ability to discriminate high quality from low quality food items (what they term 'client quality'); and the ability to forage against one's preference (to pick the lower quality food item over the preferred item; a task they term 'audience effect'). [It is important to note here that these ecological tasks involved no 'real' client or 'audience', as the task involved feeding from plexiglass plates.]

Overall, this is a very exciting study. The suggestions below should not be viewed as diminishing the importance of their results, only meant to add clarification and context.

Suggested edits:

1. Please rename the 'domain-specific' or 'ecological tasks' so they do not refer to any kind of social environment. Given that no client was present, nor any audience present during either of these tasks, those terms should not be part of their names.

2. Are these 4 tasks really a comparison of Domain-general vs Domain-specific? All 4 of them involve learning something about a foraging strategy. (a) reversal learning of a color associated with food; (b) inhibitory control of moving toward food (detour); (c) discriminating high from low quality food patch; and (d) learning to feed against preference (selecting the low quality patch when conditions are such that the high quality patch will be removed).

3. Lines 440-443. Consider editing this sentence "Females showed better inhibitory control abilities in a domain-general task (i.e., detour task), but they failed to apply these abilities to optimize their food intake in the "same" task when framed in a more ecologically-relevant context (i.e., audience effect task)." There are two aspects of this sentence that seem a bit misleading.

(a) I wouldn't use the terminology "same" task when comparing detour and 'audience effect' or 'forage against preference' task. Yes, inhibition is required in both but that is not the 'task'.

(b) In addition, the wording in the sentence has an implication that females fail but males do not fail at this specific task. Given that both sexes performed equally (poor or not) on this 'audience/forage against preference' test, I suggest changing this sentence to reflect that finding.

4. Discussion: Lines 528-529. Please omit/modify the sentence "The overall results show that not all cognitive domains are positively linked to an individual's probability to change sex." Since this study did not follow individual's across time and compare across individuals that 'changed sex' to those that did not, this statement should be removed.

Decision letter (RSOS-210239.R0)

Dear Dr Triki

On behalf of the Editors, we are pleased to inform you that your Manuscript RSOS-210239 "Sex differences in the cognitive abilities of a sex-changing fish species *Labroides dimidiatus*" has been accepted for publication in Royal Society Open Science subject to minor revision in accordance with the referees' reports. Please find the referees' comments along with any feedback from the Editors below my signature.

We invite you to respond to the comments and revise your manuscript. Below the referees' and Editors' comments (where applicable) we provide additional requirements. Final acceptance of

your manuscript is dependent on these requirements being met. We provide guidance below to help you prepare your revision.

Please submit your revised manuscript and required files (see below) no later than 7 days from today's (ie 09-Jun-2021) date. Note: the ScholarOne system will 'lock' if submission of the revision is attempted 7 or more days after the deadline. If you do not think you will be able to meet this deadline please contact the editorial office immediately.

on behalf of Kevin Padian (Subject Editor)
openscience@royalsociety.org

Associate Editor Comments to Author:
Comments to the Author:

Thank you for your patience in reviewing your work. The comments of the reviewers are largely positive, though the recommendations of tweaks would seem reasonable to add value to an already solid piece of work. Please carefully incorporate the relevant modifications.

Reviewer comments to Author:
Reviewer: 1
Comments to the Author(s)

I was one of the previous referees and I really appreciated how the authors took on board the comments provided. The authors know already how I appreciate their choice of a study system (a sex-changing fish to look at differences in sex-specific cognition capability) and their smart experiments under lab conditions.

I really like the new supplementary Figure S1, which I would add in the main text, as it is very relevant to reduce the concern of differences in cognitive abilities due to age (with size as a proxy) because of extra experience, and not to sex.

I was worried for the low sample size, but the authors responded to my concerns (to be honest, I would include a shorter version of that answer in the material and methods, to explain the ethical implication and the sex ratio skew of removing more than 10 animals from the reef),

My other concerns/suggestions have been considered.

I am still unsure why the authors write: "On the other hand, as males are successful former females, they could potentially perform above female average" Why? In what tasks? I do not see the rationale: I thought the all point was to test for inter-sexual differences in performing various cognitive tasks. I don't understand where this general statement now comes from. And why one sex should in general perform better or worse (the authors mention that the different social roles warrant different cognitive abilities).

Re-reading the MS now, I have also a fundamental question. The authors start the paper aiming to look for potential sex differences in cleaner fish cognition. But then lots of speculation is on changing sex based on cognition. It seems that the focus shifts during the course of the MS, and I am not sure the new hypothesis on sex change is really tested with the clever lab experiments performed.

I do not understand why better learners are considered more likely to change sex. According to the size advantage model, as reported in the final discussion, larger female can become the dominant males. Are the authors suggesting that a smaller female, better learner (but again, in what tasks?) than a larger female, is more likely to change sex? And are they suggesting than this smaller female would grow faster (as males are bigger than the larger female)?

If males and females excel in different task, why is the most parsimonious hypothesis that males are former females that were above-average learners? The fact that aggressive females become aggressive males (which I brought to the attention of the authors in the previous review), just seems to show that the trait is maintained. Better learner females could be better learner males, but why they would be the ones preferentially changing sex? I really thought that size (and aggression) were important, to rule the harem. But maybe I am getting this wrong. Anyway, I am not sure the experiments performed support this hypothesis, as we do not know who is changing sex (and in some cases larger females perform worse than smaller ones, even though I could not explain why and the authors comment on it, but cannot explain it either).

Indeed, the authors state in line 528-530: "The overall results show that not all cognitive domains are positively linked to an individual's probability to change sex". But why not all? And where they get the probability from? Do they find evidence for some domains? They have not tested this directly. And in the final conclusion the authors state: "more investigation is needed to see whether sex-change causes changes in cognitive performance or whether females with higher cognitive performance are those who achieve sex-change later in their life" Should not be earlier in their life, if they preferentially change sex? But I really thought the whole MS was written based on the first consideration (sex change cause changes in cognitive performance).

I am just a bit confused and I think that these two need to be better clarified.

Finally, I don't know if I would talk about intersexual conflicts... Would not be the all point that hormones would influence some abilities, so changing hormones would change them too and hormones would possibly amplify the effects on the cognitive abilities more relevant for that sex? I would not say that one cognitive ability is really detrimental to one sex: I would say it could be positive for both sexes, possibly more so in one sex than the other. But on this I am not sure.

Minor comments

Line 44: Now the introduction reads "males and females may partly differ". I made a suggestion in the previous version and I still would god for it (but it is writing style, not important: it has been suggested that may or been shown they do.

Line 57: what pathways activities?

Line 62: consider revising punctuation: reproductive strategies of the two sexes, such that females, often driven by foraging motivations, have greater cognitive flexibility, while males, driven

by finding mates, excel in spatial memory tasks

line 91: I would remove -first in both male-first and female-first) as you mention that individual reproduce first as one sex (male or female), not male-firs and female-first

Line 11: sorry, I am not familiar: do males have larger territories, and move from one cleaning station to the other? Are female cleaning stations inside the male ones?

Line 147: I would change cleaner fish into *Labroides dimidiatus*

Line 156: why intraspecific?

Line 161-162: change semicolon to comma, especially since there is

Line 172: just out of curiosity, how do you think males ill do when returned to the reef?

Line 195: TL in bracket

Line 196: I would add (mean \pm SD) the first time you see it

Line 361-362: would it be worth adding here (or in the discussion) that males do perform better (even if they are bigger)

Reviewer: 2

Comments to the Author(s)

This study takes the examination of sex differences in cognition to a new level by comparing cognitive performance between the sexes in a species that has the ability to change sex. The authors conduct a series of cognitive assays, some that examine general cognitive features (such as cognitive flexibility and inhibitory control using detour task and reversal learning assays) and others that ask an ecologically relevant cognitive task (two foraging tasks that examined an individual's ability to discriminate high quality from low quality food items (what they term 'client quality'); and the ability to forage against one's preference (to pick the lower quality food item over the preferred item; a task they term 'audience effect'). [It is important to note here that these ecological tasks involved no 'real' client or 'audience', as the task involved feeding from plexiglass plates.]

Overall, this is a very exciting study. The suggestions below should not be viewed as diminishing the importance of their results, only meant to add clarification and context.

Suggested edits:

1. Please rename the 'domain-specific' or 'ecological tasks' so they do not refer to any kind of social environment. Given that no client was present, nor any audience present during either of these tasks, those terms should not be part of their names.
2. Are these 4 tasks really a comparison of Domain-general vs Domain-specific? All 4 of them involve learning something about a foraging strategy. (a) reversal learning of a color associated with food; (b) inhibitory control of moving toward food (detour); (c) discriminating high from low quality food patch; and (d) learning to feed against preference (selecting the low quality patch when conditions are such that the high quality patch will be removed).
3. Lines 440-443. Consider editing this sentence "Females showed better inhibitory control abilities in a domain-general task (i.e., detour task), but they failed to apply these abilities to optimize their food intake in the "same" task when framed in a more ecologically-relevant context (i.e., audience effect task)." There are two aspects of this sentence that seem a bit misleading.
 - (a) I wouldn't use the terminology "same" task when comparing detour and 'audience effect' or 'forage against preference' task. Yes, inhibition is required in both but that is not the 'task'.
 - (b) In addition, the wording in the sentence has an implication that females fail but males do not fail at this specific task. Given that both sexes performed equally (poor or not) on this 'audience/forage against preference' test, I suggest changing this sentence to reflect that finding.
4. Discussion: Lines 528-529. Please omit/modify the sentence "The overall results show that not all cognitive domains are positively linked to an individual's probability to change sex." Since this study did not follow individual's across time and compare across individuals that 'changed sex' to those that did not, this statement should be removed.

===PREPARING YOUR MANUSCRIPT===

a 'clean' version of the new manuscript that incorporates the changes made, but does not highlight them. This version will be used for typesetting.
Please ensure that any equations included in the paper are editable text and not embedded images.

===PREPARING YOUR REVISION IN SCHOLARONE===

- If you are requesting a discretionary waiver for the article processing charge, the waiver form must be included at this step.
- If you are providing image files for potential cover images, please upload these at this step, and inform the editorial office you have done so. You must hold the copyright to any image provided.
- A copy of your point-by-point response to referees and Editors. This will expedite the preparation of your proof.

- Ensure that your data access statement meets the requirements at <https://royalsociety.org/journals/authors/author-guidelines/#data>. You should ensure that you cite the dataset in your reference list. If you have deposited data etc in the Dryad repository, please only include the 'For publication' link at this stage. You should remove the 'For review' link.
- If you are requesting an article processing charge waiver, you must select the relevant waiver option (if requesting a discretionary waiver, the form should have been uploaded at Step 3 'File upload' above).
- If you have uploaded ESM files, please ensure you follow the guidance at <https://royalsociety.org/journals/authors/author-guidelines/#supplementary-material> to include a suitable title and informative caption. An example of appropriate titling and captioning may be found at https://figshare.com/articles/Table_S2_from_Is_there_a_trade-off_between_peak_performance_and_performance_breadth_across_temperatures_for_aerobic_scope_in_teleost_fishes_/3843624.

Author's Response to Decision Letter for (RSOS-210239.R0)

See Appendix A.

Decision letter (RSOS-210239.R1)

Dear Dr Triki,

I am pleased to inform you that your manuscript entitled "Sex differences in the cognitive abilities of a sex-changing fish species *Labroides dimidiatus*" is now accepted for publication in Royal Society Open Science.

Please ensure that you send to the editorial office an editable version of your accepted manuscript, and individual files for each figure and table included in your manuscript. You can send these in a zip folder if more convenient. Failure to provide these files may delay the

processing of your proof. You may disregard this request if you have already provided these files to the editorial office.

on behalf of Kevin Padian (Subject Editor)
openscience@royalsociety.org

Appendix A

Associate Editor Comments to Author:

Comments to the Author:

Thank you for your patience in reviewing your work. The comments of the reviewers are largely positive, though the recommendations of tweaks would seem reasonable to add value to an already solid piece of work. Please carefully incorporate the relevant modifications.

Reply:

We thank the editor and reviewers for their feedback and comments. We have revised the manuscript accordingly, which we believe it helped improve further the clarity and flow of the manuscript considerably. Please find below our detailed replies point-by-point. Line numbers refer to the “annotated” version of the manuscript.

Reviewer comments to Author:

Reviewer: 1

Comments to the Author(s)

I was one of the previous referees and I really appreciated how the authors took on board the comments provided. The authors know already how I appreciate their choice of a study system (a sex-changing fish to look at differences in sex-specific cognition capability) and their smart experiments under lab conditions. I really like the new supplementary Figure S1, which I would add in the main text, as it is very relevant to reduce the concern of differences in cognitive abilities due to age (with size as a proxy) because of extra experience, and not to sex.

Reply:

We are very thankful for the reviewer for accepting to review our manuscript again. We hope that we have now integrated all the remaining suggestions to improve the quality and clarity of the manuscript. We have now included Figure S1 to the main text as suggested by the reviewer. It became Fig. 5 in the new revision.

I was worried for the low sample size, but the authors responded to my concerns (to be honest, I would include a shorter version of that answer in the material and methods, to explain the ethical implication and the sex ratio skew of removing more than 10 animals from the reef),

Reply :

We thank the reviewer for the suggestions. We have now added these explanations to the methods section. Line 211-217.

My other concerns/suggestions have been considered.

Reply:

We are glad to hear that.

I am still unsure why the authors write: "On the other hand, as males are successful former females, they could potentially perform above female average" Why? In what tasks? I do not see the rationale: I thought the all point was to test for inter-sexual differences in performing various cognitive tasks. I don't understand where this general statement now comes from. And why one sex should in general perform better or worse (the authors mention that the different social roles warrant different cognitive abilities).

Reply:

We agree with the referee that our previous wording was unclear, we have edited the sentence accordingly in the revision. It reads as follow:

"On the other hand, if cognition promotes growth and survival in the females, and thus sex-changing later in life, one would expect that males could potentially perform above female average". Line 164-166.

Re-reading the MS now, I have also a fundamental question. The authors start the paper aiming to look for potential sex differences in cleaner fish cognition. But then lots of speculation is on changing sex based on cognition. It seems that the focus shifts during the course of the MS, and I am not sure the new hypothesis on sex change is really tested with the clever lab experiments performed.

Reply:

We agree that our study is preliminary and it doesn't test the effect of changing sex on cognition or vice versa. However, we needed this first step to see if males and females differ in their cognitive abilities. We say this clearly in the conclusion.

I do not understand why better learners are considered more likely to change sex. According to the size advantage model, as reported in the final discussion, larger female can become the dominant males. Are the authors suggesting that a smaller female, better learner (but again, in what tasks?) than a larger female, is more likely to change sex? And are they suggesting that this smaller female would grow faster (as males are bigger than the larger female)?

Reply:

So this comment links to the previous comment about the introduction, and we hope that the clarification there solves the issue. To reiterate: we present a hypothesis to explain the results; we are not (yet) testing it. The logic is that if cognition helps a female to grow fast and to survive, then she is more likely to become a male eventually, and therefore males are on average better learners than females. Our samples cannot clarify this hypothesis, it

needs explicit testing (see concluding Remarks). The negative correlation between size and performance (in the reversal learning task) within females is only useful as support for this hypothesis in that the correlation coefficient speaks against the hypothesis that simply growing larger makes a cleaner more performant.

We also edited a sentence in the discussion linked to this matter. It reads as follow: "A second hypothesis regarding our results is that better female learners are more likely to grow fast and survive, which makes them more likely to change sex and become males. As a consequence, male learning abilities would be better than the average female ability." Line 529-532.

If males and females excel in different task, why is the most parsimonious hypothesis that males are former females that were above-average learners? The fact that aggressive females become aggressive males (which I brought to the attention of the authors in the previous review), just seems to show that the trait is maintained. Better learner females could be better learner males, but why they would be the ones preferentially changing sex?

Reply:

Please see replies above: we mention alternative hypotheses for our results. As for 'preferentially' changing sex: all individuals should be competing to grow and become males because of the higher reproductive output. So the question is who wins the competition. One possibility is that better learners do.

I really thought that size (and aggression) were important, to rule the harem. But maybe I am getting this wrong. Anyway, I am not sure the experiments performed support this hypothesis, as we do not know who is changing sex (and in some cases larger females perform worse than smaller ones, even though I could not explain why and the authors comment on it, but cannot explain it either).

Reply:

Body size is the key factor for changing sex! The question is who is most likely to grow to large sizes. We do not have a random sample in the sense that we have many medium-sized females, test their learning performance and release them to see which ones grow fastest. That will be a follow-up study. Even if there is a link, that does not exclude the possibility that few slow-growing inferior learners still reach a large body size and become a male. It is about different probabilities.

Indeed, the authors state in line 528-530: "The overall results show that not all cognitive domains are positively linked to an individual's probability to change sex". But why not all? And where they get the probability from? Do they find evidence for some domains? They have not tested this directly.

Reply:

We agree. We have edited this sentence accordingly. It reads as follow: "The overall results show that not all cognitive domains are positively linked to one sex over the other". Line 565-566.

And in the final conclusion the authors state: “more investigation is needed to see whether sex-change causes changes in cognitive performance or whether females with higher cognitive performance are those who achieve sex-change later in their life” Should not be earlier in their life, if they preferentially change sex? But I really thought the whole MS was written based on the first consideration (sex change cause changes in cognitive performance).

I am just a bit confused and I think that these two need to be better clarified.

Reply:

We have corrected this sentence. It reads as follow: “...females with higher cognitive performance are those who are more likely to achieve sex-change in their life”. Line 584.

Finally, I don't know if I would talk about intersexual conflicts... Would not be the all point that hormones would influence some abilities, so changing hormones would change them too and hormones would possibly amplify the effects on the cognitive abilities more relevant for that sex? I would not say that one cognitive ability is really detrimental to one sex: I would say it could be positive for both sexes, possibly more so in one sex than the other. But on this I am not sure.

Reply:

We wrote in the discussion the following: “The overall results show that not all cognitive domains are positively linked to one sex over the other. This insight provides avenues for studying the potential presence of sexual conflict [12] regarding cognitive abilities within the same individual first as a female, and then once it changes sex to become a male”. Line 566-569.

As you see, we are not claiming the existence of sexual conflict, but it's rather a speculation.

Minor comments

Line 44: Now the introduction reads “males and females may partly differ”. I made a suggestion in the previous version and I still would god for it (but it is writing style, not important: it has been suggested that may or been shown they do.

Reply:

We have rewritten this accordingly. It reads: “..., it has been shown that males and females do partly differ...”. Line 43.

Line 57: what pathways activities?

Reply:

Hormonal pathways. We have clarified that in the sentence. Line 57.

Line 62: consider revising punctuation: reproductive strategies of the two sexes, such that females, often driven by foraging motivations, have greater cognitive flexibility, while males, driven by finding mates, excel in spatial memory tasks

Reply:

Done.

line 91: I would remove -first in both male-first and female-first) as you mention that individual reproduce first as one sex (male or female), not male-first and female-first

Reply:

Done.

Line 11: sorry, I am not familiar: do males have larger territories, and move from one cleaning station to the other? Are female cleaning stations inside the male ones?

Reply:

That's correct, we edited the sentence a bit so it reads more clearly: "Female cleaners occupy small territories called "cleaning station" while male cleaners have a larger territory comprised of several of these females' cleaning stations". Line 114.

Line 147: I would change cleaner fish into Labroides dimidiatus

Reply:

Done.

Line 156: why intraspecific?

Reply:

Because we do not see much differences at the level of interspecific social interactions with client fish. Also, we talk about intraspecific because the male has more social interactions with a harem while females take care of their relatively smaller territory and interact most of the time with the male alone.

Line 161-162: change semicolon to comma, especially since there is

Reply:

Done.

Line 172: just out of curiosity, how do you think males will do when returned to the reef?

Reply:

We actually don't know. Data on females show that they reintegrate very well. But the referee is right in pointing out that the dynamics should differ for males. We will keep this in mind for future studies. Research by Japanese colleagues has shown that cleaners can change sex more than once. Potentially, the newly emerged males might hence reverse. Alternatively, harems will become smaller. Collecting data on that should be rather straightforward.

Line 195: TL in bracket

Reply:

Done.

Line 196: I would add (mean \pm SD) the first time you say it

Reply:

Done.

Line 361-362: would it be worth adding here (or in the discussion) that male do perform better (even if they are bigger)

Reply:

Line 451-452, we added this to the discussion now. It reads: "... Nevertheless, the males did outperform the females in this task despite being larger in body size."

Reviewer: 2

Comments to the Author(s)

This study takes the examination of sex differences in cognition to a new level by comparing cognitive performance between the sexes in a species that has the ability to change sex. The authors conduct a series of cognitive assays, some that examine general cognitive features (such as cognitive flexibility and inhibitory control using detour task and reversal learning assays) and others that ask an ecologically relevant cognitive task (two foraging tasks that examined an individual's ability to discriminate high quality from low quality food items (what they term 'client quality'); and the ability to forage against one's preference (to pick the lower quality food item over the preferred item; a task they term 'audience effect'). [It is important to note here that these ecological tasks involved no 'real' client or 'audience', as the task involved feeding from plexiglass plates.]

Overall, this is a very exciting study. The suggestions below should not be viewed as diminishing the importance of their results, only meant to add clarification and context.

Reply:

We thank the reviewer for their efforts in providing comments on our manuscript please see below for our replies to each point in turn. We hope that we succeeded in addressing all the concerns raised in these comments.

Suggested edits:

1. Please rename the ‘domain-specific’ or ‘ecological tasks’ so they do not refer to any kind of social environment. Given that no client was present, nor any audience present during either of these tasks, those terms should not be part of their names.

Reply:

We absolutely agree that this can be misleading. We, however, added a brief explanation to the methods. It reads as follow: “It is important to note here that in all the cognitive tasks we perform in this study, we used Plexiglas plates with food as surrogates for client fish. The tasks did not involve “real” clients at any stage.” Line 228-230.

We also added quotation marks when we say “audience effect” and “client quality” throughout the text. Hope this made it more clear that we are not using real clients throughout the cognitive tasks. We would like to keep these terminology to keep a logic continuation between publications from our lab where we have used “audience effect task” with Plexiglas plates repeatedly. For instance, in the studies [1–3].

2. Are these 4 tasks really a comparison of Domain-general vs Domain-specific? All 4 of them involve learning something about a foraging strategy. (a) reversal learning of a color associated with food; (b) inhibitory control of moving toward food (detour); (c) discriminating high from low quality food patch; and (d) learning to feed against preference (selecting the low quality patch when conditions are such that the high quality patch will be removed).

Reply:

Most cognitive experiments in animals involve food as reward. Hence also the literature on domain general abilities in primates and other endotherm vertebrates involve food as reward. The difference between domain-general and domain specific is hence whether the problem is presented in an ecologically relevant way or in an abstract way. Reversal learning in the context foraging does not happen to cleaner fish in nature: a large parrotfish will always offer more food than a small damselfish. That’s why the reversal learning experiment does not tap into existing foraging strategies. Similarly, detouring transparent barriers is not something cleaners ever face in nature, while feeding against preference is their daily life.

3. Lines 440-443. Consider editing this sentence “Females showed better inhibitory control abilities in a domain-general task (i.e., detour task), but they failed to apply these abilities to optimize their food intake in the “same” task when framed in a more ecologically-relevant context (i.e., audience effect task).” There are two aspects of this sentence that seem a bit misleading.

(a) I wouldn't use the terminology “same” task when comparing detour and ‘audience effect’ or ‘forage against preference’ task. Yes, inhibition is required in both but that is not the ‘task’.

(b) In addition, the wording in the sentence has an implication that females fail but males do not fail at this specific task. Given that both sexes performed equally (poor or not) on this ‘audience/forage against preference’ test, I suggest changing this sentence to reflect that finding.

We have corrected this sentence accordingly, it reads as follow: “Females showed better inhibitory control abilities in a domain-general task (i.e., detour task) than males. However, they failed to outperform males by applying these inhibitory control abilities to optimise their food intake in a more ecologically-relevant context, i.e. feeding against preference (“audience effect” task).” Line 438-441.

4. Discussion: Lines 528-529. Please omit/modify the sentence “The overall results show that not all cognitive domains are positively linked to an individual's probability to change sex.” Since this study did not follow individual's across time and compare across individuals that ‘changed sex’ to those that did not, this statement should be removed.

Reply:

We totally agree. Similar concerns were raised by Reviewer#1, the sentence now reads as follow in the new revision: “The overall results show that not all cognitive domains are positively linked to one sex over the other”. Line 565-566.

Cited literature:

1. Bshary R, Grutter AS. 2006 Image scoring and cooperation in a cleaner fish mutualism. *Nature* **441**, 975–978. (doi:10.1038/nature04755)
2. Triki Z, Wismer S, Rey O, Ann Binning S, Levorato E, Bshary R. 2019 Biological market effects predict cleaner fish strategic sophistication. *Behav. Ecol.* **30**, 1548–1557. (doi:10.1093/beheco/arz111)
3. Triki Z, Wismer S, Levorato E, Bshary R. 2018 A decrease in the abundance and strategic sophistication of cleaner fish after environmental perturbations. *Glob. Change Biol.* **24**, 481–489. (doi:10.1111/gcb.13943)